# Chronic Stress Detrimentally Affects In Vivo Maturation in Rat Oocytes and Oocyte Viability at All Phases of the Estrous Cycle

**DOI:** 10.3390/ani11092478

**Published:** 2021-08-24

**Authors:** Fahiel Casillas, Miguel Betancourt, Lizbeth Juárez-Rojas, Yvonne Ducolomb, Alma López, Alejandra Ávila-Quintero, Jimena Zamora, Mohammad Mehdi Ommati, Socorro Retana-Márquez

**Affiliations:** 1Department of Biology of Reproduction, Iztapalapa Campus, Metropolitan Autonomous University, Mexico City 09340, Mexico; fahiel@xanum.uam.mx (F.C.); alizjuarezroj@gmail.com (L.J.-R.); alejandra.avila.quintero@gmail.com (A.Á.-Q.); jimenaogz4@gmail.com (J.Z.); 2Department of Health Sciences, Iztapalapa Campus, Metropolitan Autonomous University, Mexico City 09340, Mexico; bet@xanum.uam.mx (M.B.); duco@xanum.uam.mx (Y.D.); almaxim@hotmail.com (A.L.); 3Department of Bioinformatics, College of Life Sciences, Shanxi Agricultural University, Jinzhong 030801, China; mehdi.ommati@gmail.com

**Keywords:** cold stress, female cycles, homeostasis, oogenesis, ovum

## Abstract

**Simple Summary:**

Recently, a significant relationship between stress and reproductive failure in women was reported; being one of the possible causes of infertility. The World Health Organization recognizes infertility as a global public health issue; therefore, the interest in understanding the main causes of this issue has increased over the last few decades. Thus, many studies have reported that stress can adversely alter the functionality of the hypothalamic-pituitary-gonadal axis; as well as being one of the reasons of subfertility in patients undergoing in vitro fertilization. Therefore, it can be assumed that stress is closely related to poor in vitro fertilization outcomes. In chronically stressed female rats, irregular estrous cyclicity, increased corticosterone levels, decreased oocyte viability, and increased percentage of abnormal oocytes were obtained in all estrous cycle phases, resulting in reduced oocyte maturation during proestrus. Oocyte maturation disturbed by chronic stress is a crucial factor by which chronic stress disrupts female reproduction.

**Abstract:**

Background: Stress has been considered as one of the causes of decreased reproductive function in women. However, direct evidence of the effect of chronic stress on oocytes depending on estrous cycle phases is limited. Objective: The present study aimed to evaluate the impact of chronic stress on the viability, integrity, and maturation of rat oocytes depending on estrous cycle phases, specifically proestrus, estrus, and diestrus. Methods: For this purpose, adult female rats were stressed daily by cold water immersion (15 °C) for 30 consecutive days. Results: In chronically stressed female rats, irregular estrous cyclicity, increased corticosterone levels, decreased oocyte viability, and an increased percentage of abnormal oocytes were obtained in all the estrous cycle phases, resulting in reduced oocyte maturation during proestrus. Conclusion: Oocyte maturation disturbed by chronic stress is a crucial factor by which chronic stress disrupts female reproduction

## 1. Introduction

Stress is defined as a disruption in an organism’s homeostasis, and the stress response depends on the intensity and duration of the stimuli (stressor). Stress, both acute or chronic forms, can result in physical or physiological disorders. Recently, a significant relationship between stress and reproductive failure in women was reported [1,2,3]; being one of the possible causes of infertility. The World Health Organization recognizes infertility as a global public health issue; therefore, the interest in understanding the main causes of this issue has increased over the last few decades. Thus, many studies have reported that stress can adversely alter the functionality of the hypothalamic-pituitary-gonadal (HPG) axis, as well as being one of the reasons of subfertility in patients undergoing in vitro fertilization (IVF) [4,5]. Therefore, it can be assumed that stress is closely related to poor IVF outcomes.

Stressors cause the over-activation of the hypothalamic-pituitary-adrenal (HPA) axis and, subsequently, increase cortisol (in humans) or corticosterone (in rodents) secretion. The activation of this axis, both acutely or chronically, has been shown to impair female reproduction, directly via hypothalamic, pituitary, ovarian, and uterine levels, or indirectly through neuroendocrine related routes [3]. In female rats, chronic unpredictable mild stress lengthens the estrous cycle to 6–7 days, mainly due to prolonged diestrus [6]. Furthermore, in female rodents, stress disrupts ovarian follicular development [7], ovulation [8], and diminishes oocyte quality and embryo development [9]. Activation of the HPA axis increases cortisol levels and the consequent oxidative stress, affecting granulosa cell functions leading to apoptosis and reducing their ability to synthesize estradiol (E_2_), which, subsequently, impairs the normal development of human oocytes [5]. Furthermore, glucocorticoids administered to female mice induce apoptosis in mural granulosa cells and increase Fas ligands in cumulus cells and oocytes, impairing oocyte competence by triggering apoptosis [10]. Chronic restraint stress can also cause the immoderate activation of ovarian primordial follicles [11] and repeated restraint stress impairs oocyte development, through cumulative effects on growing ovarian follicles, with antral follicles being more sensitive to stress [12].

A study of the effects produced by stress on female reproduction is difficult to achieve because the precise determination of the estrous cycle phases is required; also, the fluctuating sex hormones per phase increases the number of animals used [13]. Hormonal variations, depending on the estrous cycle phase, make experimental work more complex. In one study evaluating the effects of stress on rodent oocytes, cells were collected regardless of the estrous cycle phase, then matured in vitro, showing impaired oocyte development potential due to chronic stress [14]. Other studies stimulate mice via hormone with pregnant mare serum gonadotropin (PMSG) [15] or equine chorionic gonadotropin (eCG) [12] to collect matured oocytes. As far as we know, to date, there are no studies evaluating the effects of chronic stress on oocytes collected in the different phases of the estrous cycle, without any exogenous hormonal stimulation, which can give more accurate information about how stress affects their development. Therefore, the present study aimed to evaluate the effect of chronic stress on the viability, integrity, and in vivo maturation of rat oocytes depending on three estrous cycle phases, including proestrus, estrus, and diestrus.

## 2. Materials and Methods

### 2.1. Ethics

The present study was approved by the Health and Biological Sciences Division Ethics Committee at Autonomous Metropolitan University-Iztapalapa Campus. Animal management and experiments were performed under the official Mexican regulations (NOM-062-ZOO-1999) and the domestic and laboratory animal regulations published in the Ethical Conduct Guidelines for Research, Teaching, and Outreach of the Health and Biological Sciences Division.

### 2.2. Animals

Nighty healthy mature Female Wistar rats (2.5 months old, 200 g live bodyweight) were obtained from the Autonomous Metropolitan University vivarium. Females were assigned to control (*n* = 45) or stress (*n* = 45) groups. Animals were maintained in acrylic cages (10 per cage), with an inverted light/dark cycle (12/12; lights off at 09:00 am), and food and water were ad libitum. To ensure proper cycling in all females, daily vaginal smears were obtained and assessed for two weeks before experiment [6].

### 2.3. Experimental Design

To evaluate the effects of chronic stress by cold water immersion on oocyte viability, integrity, and maturation, female rats were stressed for 30 consecutive days, and vaginal cytology was evaluated before being sacrificed. Oocyte viability, integrity, and maturation, as well as estrous cyclicity and serum corticosterone, were assessed at the end of stress exposure.

### 2.4. Stress Procedure

Control females were left undisturbed in their cages. The females from the stress group were transferred to another room, then placed individually in a covered tank with cold water (temperature = 15 °C; depth = 15.5 cm) for 15 min. Rats remained in an upright position, keeping their head above water level [16]. At the end of the 15 min, rats were picked up from the tank and towel dried. The stressor was applied once a day, when lights were off, at 09:00 am, for 30 consecutive days.

### 2.5. Vaginal Cytology and Estrous Cyclicity

Estrous cycles were evaluated using vaginal smears in all females from the control and stress groups for 30 days (Figure 1). The smears were obtained one hour before the dark period’s onset, (08:00 am), using a stainless-steel loop (2 mm id) with saline, then mounted on a glass slide and stained with hematoxylin-eosin and evaluated under a light microscope (Olympus, model CX41RF, Shinjuku-Ku, Tokyo, Japan) at 200× and 400× magnifications. Estrous cycle phases were identified according to vaginal cytology: proestrus, nucleated cells; estrus, cornified cells; diestrus 1, cornified cells, and leucocytes; diestrus 2, leucocytes [6].

### 2.6. Biological Samples

After 30 days of stress, vaginal smears were obtained to identify the estrous cycle phases. Then, females were rapidly euthanized, and ovaries were obtained from females in proestrus (control *n* = 15; stress *n* = 15), estrus (control *n* = 15; stress *n* = 15), and diestrus (control *n* = 15; stress *n* = 15). In the rat, the metestrus phase is also known as diestrus 1, in which hormone levels are similar to those of the diestrus. The transition from diestrus 1 to diestrus 2 is very fast, which makes it difficult to find enough females in this phase. For this reason, we do not include the metestrus in further evaluations. The total number of evaluated oocytes was proestrus (control *n* = 185; stress *n* = 191), estrus (control *n* = 48; stress *n* = 46), and diestrus (control *n* = 147; stress *n* = 144). Trunk blood was also collected (2–3 mL) from each rat for corticosterone assay.

### 2.7. Hormonal Analysis

Corticosterone was extracted from serum and quantified using high performance liquid chromatography (HPLC) [17]. Briefly, 100 µL of 19-nortestosterone solution (5 mg/mL in methanol/water, 60:40, *v*/*v*), as an internal standard, were added to serum (1 mL) and mixed. Corticosterone was extracted into 5 mL of diethyl ether:dichloromethane (60:40, *v*/*v*) by vortex mixing and immediately centrifuged (1050 g, 4 °C, 5 min). The organic phase was obtained, vortex mixed with 1 mL of HPLC-grade water, and centrifuged. Afterward, the organic phase was obtained (3 mL); after evaporation at room temperature, the residue was re-dissolved in 100 mL of methanol:water (60:40, *v*/*v*). Chromatographic system [17]. The guard column (Symmetry C18, particle size 3.5 mm, 2.1 mm × 10 mm; Waters Corp., Milford, MA, USA) and the column were equilibrated using HPLC-grade water: acetonitrile (65:35, *v*/*v*) at a flow rate of 0.4 mL/min. Separations were made at 40 °C in a Waters Symmetry C18 column (2.0 mm × 150 mm; particle size 5 mm; Waters Corp., Milford, MA, USA). A Waters 600-MS system controller was used for flushing the mobile phase, and the steroids were assessed using a 486 Water UV absorbance detector (fitted at 250 nm). Finally, the obtained results were analyzed using the Millennium 32 software (Waters Corp., Milford, MA, USA). The detection limit of the assay for corticosterone was 5 ng/mL.

### 2.8. Ovarian Oocyte Recovery

Unless otherwise stated, all chemicals were purchased from Sigma Chemical Co. (St. Louis, MO, USA). For cumulus-oocyte complexes’ (COCs) collection, ovaries from control and stressed rats in proestrus, estrus, and diestrus were collected in a Petri dish containing Tyrode modified medium supplemented with 10 mM sodium lactate, 10 mM HEPES, and 1 mg/mL polyvinyl alcohol (PVA) (TL-HEPES-PVA) at pH 7.3–7.4 tempered to 37 °C. COCs were collected in the different phases of the estrous cycle; at the time of collection, the stage of maturation of the oocytes was evaluated to ensure that induction of maturation was not performed in vitro nor by hormonal stimulation. To obtain the COCs, all the ovarian follicles were torn using two insulin needles (8 mm). All observed COCs were collected. For evaluation, approximately 30–40 COCs per group and estrous cycle phase were placed in each well of a four-well dish (Thermo-Scientific Nunc, Rochester NY, USA) containing 100 μL of tissue culture media (TCM-199) and 0.1% hyaluronidase for less than 5 min for denudation. Then, oocytes were transferred to a TCM-199 hyaluronidase-free media for evaluation.

### 2.9. Evaluation of Oocyte Viability

Viability was measured immediately after oocyte denudation (T0). Oocytes were added to a 100-microliter drop of 0.5 mg/mL methyl-thiazolyl-tetrazolium (MTT) diluted in phosphate-buffered saline solution (PBS) and incubated at 37 °C with 5% CO_2_ in air and humidity at saturation [18]. After 30 min, oocytes were analyzed under a light microscope (Zeiss Axiostar, Germany) and classified as non-viable (colorless) (Figure 2(Aa)) or viable cells (with purple coloration) (Figure 2(Ab)). The MTT stain evaluates the metabolic activity of cells. NADPH-dependent cellular oxidoreductase enzymes can reduce the tetrazolium dye MTT to formazan (reflecting purple coloration) as an indicator of viable cells.

### 2.10. Evaluation of Oocyte Maturation

After the removal of cumulus cells, under an inverted microscope using the bright field, the oocyte maturation stages were evaluated (Figure 3(Aa–c)). Maturation was also evaluated using the Hoechst stain. Oocytes were exposed to 10 μg/mL of Hoechst 33342 for 40 min. Oocyte evaluation was performed using a confocal scanning laser microscope (Zeiss, LSM T-PTM, Germany) (Figure 3(Aa’–c’)). Oocytes in germinal vesicle (GV; Figure 3(Aa,a’)) and metaphase-I (MI; Figure 3(Ab,b’)) were considered as immature, and those in metaphase-II (MII) with the presence of the first polar body as mature (Figure 3(Ac,c’)).

### 2.11. Statistical Analysis

The number of estrous cycles and estrous phases was analyzed using Student’s *t*-test. Corticosterone serum level concentrations were analyzed using two-way ANOVA (condition and stages as factors), followed by the Newman–Keuls post hoc test. Percentage of oocyte viability, in vivo maturation, and abnormal oocytes from control and stressed females were analyzed using the chi-square test followed by post hoc Fisher’s exact test. Differences were considered significant when *p* < 0.05, and data are expressed as Mean ± SEM. Data analyses were performed using Prism 8 for macOS, version 8.2.1 software.

## 3. Results

### 3.1. Estrous Cyclicity

The vaginal smears from the control and stress females showed differences in their estrous cycle. All the control females presented normal estrous cycles with a duration of 4–5 days and a normal progression of all the phases. The average number of estrous cycles in 30 days was 7.1 ± 0.17 in the control group. Stressed females showed irregular and prolonged estrous cycles, increasing the number of consecutive days (3 days) in proestrus, estrus, and diestrus (Figure 1). The average number of estrous cycles in 30 days was 3.1 ± 0.34 (t = 10.21, *p* < 0.0001).

### 3.2. Oocyte Viability

When the effect of chronic stress on oocyte viability was evaluated, we found that it was significantly reduced in all the estrous cycle phases (44.6 ± 8.81% proestrus; 28.6 ± 14.89% estrus; 28.6 ± 10.47% diestrus) compared to the control (85.6 ± 3.48% proestrus; 73 ± 4.50% estrus; 78.3 ± 5.48% diestrus) (Figure 2B; *p* < 0.0001).

### 3.3. In Vivo Oocyte Maturation

Regarding oocyte in vivo maturation, the percentage of oocytes that reached the MII stage decreased up to 7.3 ± 3.48% in the stressed females during proestrus, compared to 52 ± 9.83% in the control (Figure 3B; *p* < 0.0001); however, no significant differences were observed during estrus and diestrus at the MII stage. Accordingly, the percentage of oocytes at the germinal vesicle (GV) stage increased in stressed females during proestrus 46.3 ± 11.26% and diestrus 64.3 ± 9.81% compared to those in the control groups (17.2 ± 1.85% and 51.8 ±1%, respectively). However, no significant differences were observed during estrus at the GV stage. The percentage of oocytes that reached the MI stage decreased significantly during estrus 29.3 ± 8.43% and diestrus 23 ± 12.41% compared to the control females (39.4 ± 5.03 and 37± 11.26%, respectively) (Figure 3B; *p* < 0.05).

### 3.4. Abnormal Oocytes after Chronic Stress

The percentage of abnormal oocytes was higher in the stressed females in proestrus 5.3 ± 0.33%, estrus 18 ± 2%, and diestrus 12 ± 6.65% than in the control rats (0.6 ± 0.33% proestrus; 2.3 ± 0.33 diestrus) (Figure 4B; *p* < 0.05). Some of the abnormalities observed in the oocytes are shown in Figure 4A, such as zona pellucida deformation and rupture, compact cytoplasm (Figure 4(Aa)), abnormal MII-oocyte zona pellucida and plasma membrane, as well as vacuolated-granular cytoplasm (Figure 4(Ab)). Additionally, 30% of the oocytes in GV showed an eccentric GV in the stressed females. The normal GV position is marked with an arrow (Figure 4(Ac)).

### 3.5. Corticosterone Serum Levels and Oocyte Viability after Chronic Stress

In the control females, the corticosterone serum levels remained in a range of 250–300 ng/mL during the estrous cycle phases. In the stressed females, corticosterone increased to 600–620 ng/mL in all the estrous cycle phases compared to the control (*p* < 0.0001; Figure 5).

## 4. Discussion

Different types of chronic stress have been widely associated with alterations in female reproductive function; however, the specific intraorganellar mechanisms in vivo, by which it may interfere with reduced fertility in terms of oocyte viability, integrity, and maturation are limited. An evaluation of the mechanisms of damage produced by chronic stress in oocytes depending on the estrous cycle phases allows for a better understanding of its possible relationship with infertility.

### 4.1. Estrous Cyclicity

The results showed considerable alterations in the estrous cycle of the stressed rats; the females remained in the diestrus and estrus phases for up to three consecutive days, lengthening their estrous cycle duration. This effect has been previously reported in other studies using different stress paradigms [6,19,20]. This effect could be explained because chronic stress can produce alterations in the levels of E_2_ during proestrus and diestrus and luteinizing hormone (LH) during proestrus, affecting the ovulation rate and fertility [20]. In the present study, the results of the measurement of steroid hormone levels were not included because they were already published in a previous study by the research group [20]. We found that in the stressed rats, E_2_ levels decreased significantly during proestrus and diestrus (40 pg/mL and 25 pg/mL, respectively) compared to the control (83 pg/mL and 50 pg/mL, respectively). In addition, we found that the progesterone levels in the stressed rats decreased significantly during diestrus (12 ng/mL) compared to the control (22 ng/mL). Therefore, with these results, we set out to evaluate the effect of chronic stress in oocytes since, if during the proestrus phase there is a decrease in E_2_, this may have an impact at the follicular level, affecting oocyte maturation.

### 4.2. Oocyte Viability and Maturation after Chronic Stress

In the present study, we observed that in female rats chronically stressed by cold water immersion (15 °C), corticosterone levels increased significantly in all the phases of the estrous cycle compared with that in the control group. Additionally, in stressed females, oocyte viability significantly decreased in all the estrous cycle phases compared with the control rats. In this regard, we previously reported a decrement in the hypothalamic content of Kisspeptin and gonadotropin-releasing hormone (GnRH), reduced LH levels, and consequently, a reduction in the serum concentrations of E_2_ and progesterone in chronically stressed female rats [20]. The reproductive function is regulated by the HPG axis via their crucial hormones. These hormones are responsible for regulating reproductive function through positive or negative feedback. The hypothalamic release of GnRH promotes the secretion of LH and follicle-stimulating hormone (FSH) by the pituitary. These gonadotropins act in the ovary, promoting steroidogenesis by the theca and mural granulosa cells for oocyte growth and maturation. Several studies have reported that neuropeptides, such as Kisspeptin and RFamide-related peptide-3 (RFRP-3), regulate the release of GnRH in mammals [21]. Kisspeptin neurons express both E_2_ and androgen receptors, stimulating GnRH release [22]. Quite the opposite, RFRP-3 inhibits Kisspeptin, GnRH, and LH secretion [23] in rats [24], adversely affecting the reproductive axis functionality in females [21]. During stress, the HPA axis is activated, and glucocorticoids are synthesized. Corticotropin-releasing hormone (CRH) stimulates adrenocorticotropic hormone (ACTH) release from the pituitary, which stimulates glucocorticoid synthesis. These steroid hormones reach the ovary through the bloodstream, exerting adverse effects on ovarian cells [25]. Additionally, glucocorticoids can also stimulate RFRP-3 expression in the hypothalamus [26], thus contributing to the inhibitory effects of the HPA axis on the HPG axis.

As mentioned above, Kisspeptin is involved in the onset of puberty, as well as in the regulation of LH and FSH secretion during the estrous cycle. However, in stressed female rats, Kisspeptin decreases in the anteroventral periventricular nucleus (AVPV) during proestrus and diestrus, decreasing the GnRH content in female rats [20]. It was reported that the administration of Kisspeptin (10 μg/mL) during sheep oocyte in vitro maturation has been shown to increase maturation rates [27]. Additionally, the effect of Kisspeptin on bovine granulosa cell viability was evaluated using the MTT assay. It was reported that Kisspeptin at a concentration of 100 nM decreases granulosa cell viability [28]. In addition, RFRP-3 has been reported to decrease the number of estrous cycles in cold-stressed rats [29], and to decrease follicular viability in cats [30]. Follicles’ exposure to 1 μM of RFRP-3 increased the proportion of follicles with cell death. One of the mechanisms proposed by the authors is that RFRP-3 can promote follicular degradation through paracrine signaling [30]. Additionally, RFRP-3 has been reported to decrease porcine granulosa cell viability at different concentrations [31]. Granulosa cells are critical for the maintenance of oocyte viability and functionality given the communication they establish throughout their development. If granulosa cells die, the developmental potential of oocytes could be compromised. However, little is known about the direct effect of RFRP-3 on gametes. With the results obtained in the present study, we can speculate that decreased Kisspeptin and GnRH levels, and RFRP-3 secretion during stress can adversely affect all the components of the follicle, including granulosa cells and oocytes through paracrine signaling, affecting their metabolic activity, leading to cell death.

In the present study, when evaluating oocyte maturation rates, we found that during proestrus, chronic stress significantly decreased the meiotic progression of oocytes to the MII stage, and most of them remained arrested at the GV stage. In the control animals, during proestrus, a higher percentage of oocytes were obtained in MII, which would be ovulated during estrus [32]. As mentioned earlier, the oocyte maturation process involves the activation of the HPG axis for the synthesis of GnRH, LH, FSH, and E_2_. During follicular development and oogenesis, gonadotropins have receptors on theca and mural granulosa cells to synthesize steroid hormones. The cumulus granulosa cells surrounding the oocyte in response to gonadotropins, mainly by the action of LH, promote meiotic resumption [33]. The preovulatory LH surge, in response to GnRH during proestrus, triggers the meiotic resumption of the prophase of meiosis I, known as GV breakdown, completing meiosis I. Then, the oocytes in antral follicles enter meiosis II and arrest at MII until fertilization [34]. Considering that chronic stress can disrupt GnRH and LH [20], crucial signaling hormones for oocyte meiotic resumption, it was essential to evaluate the effect of stress on oocytes during the estrous cycle to provide a better understanding of the damage produced by chronic stress in female reproduction. The decrease in oocyte viability and maturation caused by chronic stress could also explain the low fertility rates reported in stressed females [20,35]. The reduction in the percentage of MII oocytes observed in the present study could be due to a disruption in hormone production due to corticosterone, acting at hypothalamic, pituitary, and ovarian levels, causing low viability and oocyte maturation. Corticosterone could directly affect oocyte development potential, as shown in some studies in which cortisol administration in female mice caused a decrease in oocyte development potential and oocyte quality, increasing apoptosis in oocytes and mural granulosa cells via the Fas system, together with low levels of LH and E_2_ [10].

On the other hand, the percentage of oocytes in MI was lower in chronically stressed rats during estrus and diestrus than in the controls. However, in proestrus and diestrus, a higher percentage of oocytes in GV was observed as compared with that in the control group. It was recently reported that chronic stress by maternal separation increased the reactive oxygen species (ROS) levels in oocytes, affecting the development of subsequent in vitro embryos [9]. In this regard, it has been reported that increased cortisol and ROS levels result in oxidative stress-inducing granulosa cell apoptosis, decreasing E_2_ biosynthesis, and finally inducing oocyte apoptosis [5]. To our knowledge, an evaluation of oocyte viability and maturation in chronically stressed female rats by cold water immersion depending on the estrous cycle phases has not been reported. However, to support our findings, a study reported that the in vitro exposure of mouse oocytes to increased concentrations of corticosterone (0.25–250 µM) decreased the percentage of MII oocytes (45.3%) compared with the control group (86.3%). They also reported that the fertilization and embryo development rates were significantly decreased [36]. Recently, Dehdehi and colleagues (2020) [9] have reported that maternal separation-induced early life chronic stress in mice decreased the oocyte quality and embryo development in vitro. They have also shown that chronic stress reduces oocyte in vivo maturation (induced by PMSG and hCG stimulation) down to 49.58% (MII). In the present study, we found that chronic stress by cold water immersion exerted a more significant negative effect than other types of stressors since in vivo maturation, without any hormonal stimulation, was reduced by up to 7% during proestrus.

It is also well-known that temperature alterations, either hot [37] or cold [11], cause adverse effects on the reproductive system. It has recently been reported that cold stress (0–1 °C) lengthened the estrous cycle in rats and reduced testosterone, E_2_, and progesterone levels, leading to reproductive organ dysfunction [38]. Additionally, restraint stress caused apoptosis in cumulus cells, reducing the oocyte developmental potential to the blastocyst stage [39]. Another study reported that in female Sprague Dawley rats, exposure to cold (−10 °C) reduced weight gain, lengthened the estrus and diestrus phases of the estrous cycle, decreased progesterone levels, and increased LH levels [11]. Furthermore, this study also reported a decrease in the cell layer diameter of theca and granulosa cells in cold-exposed rats. However, they did not evaluate the oocyte integrity in post-cold stress; hence, further studies are needed to assay this hypothesis.

### 4.3. Abnormal Oocytes after Chronic Stress

The present study showed that in the stressed females, a higher percentage of abnormal oocytes is observed compared to the control rats. The percentage of abnormal oocytes in proestrus, estrus, and diestrus was 6, 18, and 12%, respectively. To our knowledge, this is the first study evaluating the effect of chronic stress by cold water immersion on the morphology and in vivo maturation of rat oocytes recovered in the different estrous cycle phases. More studies have also reported that stress can cause a lengthening of the estrous cycle, increasing the number of days in estrus and diestrus [11,20]. In the present study, a higher percentage of abnormal oocytes was observed in these phases. This fact suggests that oocytes could suffer morphological damage because they remain longer in an inadequate follicular hormonal environment. Accordingly, we found that in the stressed females during estrus and diestrus, a reduced percentage of MI oocytes and a high percentage of GV oocytes during diestrus are observed. Moreover, oocyte viability was compromised in both phases, and oocytes with z deformed or broken zona pellucida, an abnormal plasma membrane, and a retracted, granular, and vacuolated cytoplasm, without a regular spherical shape, were observed. Additionally, the GV position was eccentric in 30% of the oocytes evaluated in the stressed females. It has been shown that central GV oocytes develop to blastocyst with a higher frequency than eccentric GV oocytes [40]. In support of our findings, it was reported that chronic stress increases ROS levels and induces meiotic spindle abnormalities, chromatin misalignment, and mitochondrial dysfunction in mice oocytes [15]. However, melatonin optimized culture systems are able to mitigate these adverse effects in vitro [15]. Another study reported that chronic restraint stress causes oxidative stress and apoptosis in antral follicles and reduces blastocyst embryos and live pups obtained from transferred embryos [12].

In oocytes, ROS formation is notably increased in response to various conditions, including stress [15]. Although during follicular development, cumulus cells protect the oocytes against ROS-induced damage, it is known that chronic stress is capable of inducing apoptosis in these cells. In-depth reports have shown that ROS can diffuse and pass through the cell membrane, producing damage to nucleic acids, proteins, and lipids, leading to the production of abnormal oocytes with low developmental potential [41]. Normal cytoskeleton and mitochondria distribution, chromatin organization, and central GV [40] are essential markers to predict oocyte developmental competence. In the present study, cytoskeleton and chromatin organization were not assessed; therefore, studies on these parameters in oocytes from stressed females are necessary to elucidate its competence. The low competence of oocytes due to stress has been studied. It has been shown that stress for 24 h increased the CRH levels in the serum, ovaries, and oocytes, inducing apoptosis in the mural granulosa cells, reducing oocyte ability to develop as an embryo [37]. The findings reported in the present study are essential for understanding the intraorganellar mechanisms by which chronic stress alters female reproduction. In future studies, we propose to evaluate the fertilizing capacity of oocytes from chronic stressed females.

## 5. Conclusions

Stress has been linked to fertility impairment. This study showed that chronic stress affects oocyte developmental potential in the estrous cycle phases. Additionally, chronic stress disrupted estrous cyclicity, decreased oocyte viability, and increased abnormal oocyte production in all the estrous cycle phases, which resulted in a reduced oocyte maturation during proestrus. These results highlight some of the alterations produced in oocytes by which stress alters female reproduction.

## Figures and Tables

**Figure 1 animals-11-02478-f001:**
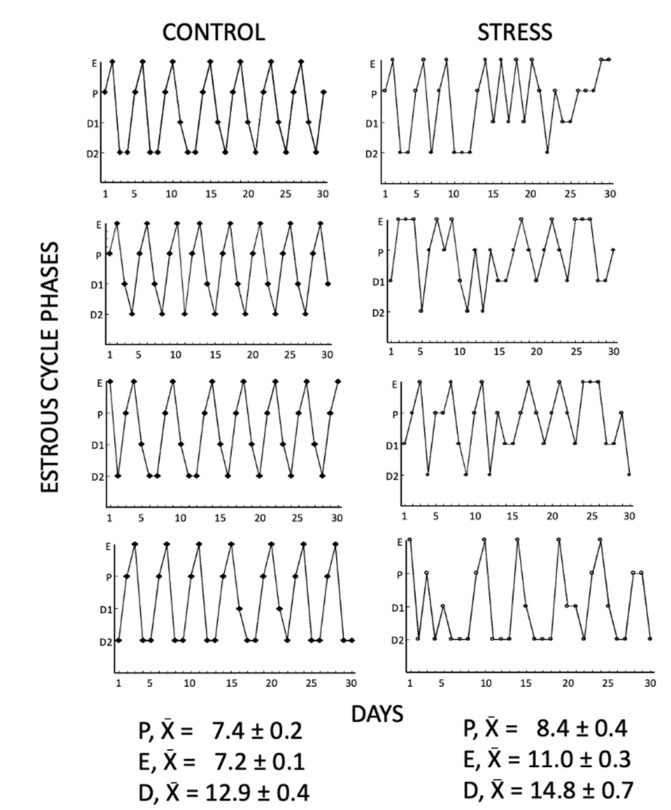
Representative estrous cycles of 4 control and 4 stressed females for 30 days. Chronic stress caused irregular and prolonged estrous cycles, increasing the number of consecutive days (3) in proestrus, estrus, and diestrus compared to control females. E= estrus, P = proestrus, D1= diestrus 1, D2 = diestrus 2.

**Figure 2 animals-11-02478-f002:**
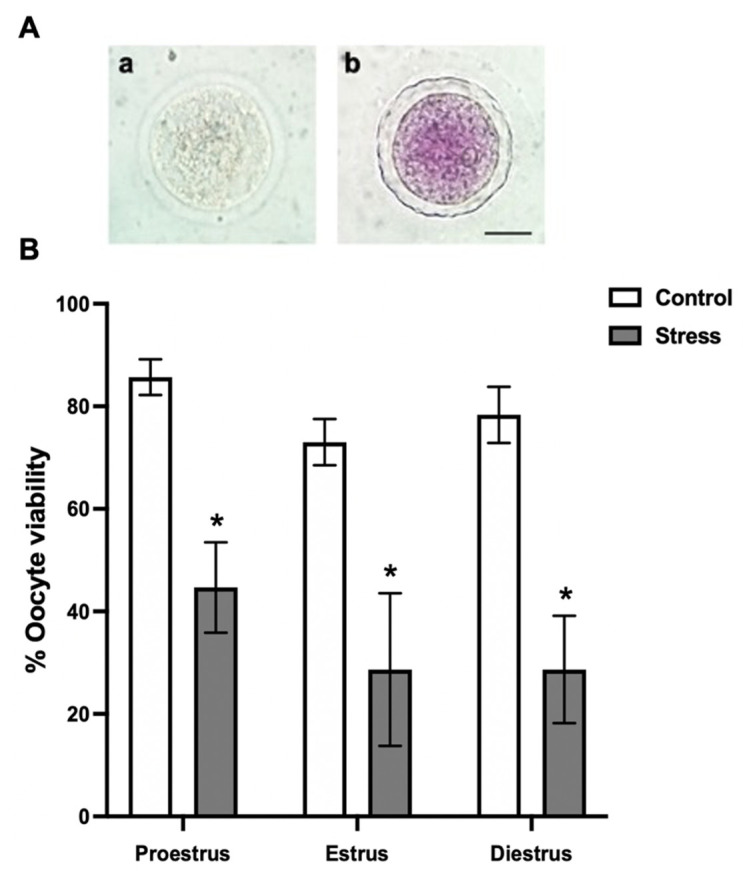
Oocyte viability depending on the estrous cycle phases in control and stressed adult female rats. (**A**) Representative images from oocyte viability evaluation by the MTT-stain: (**a**) colorless dead oocyte and (**b**) purple-stained alive oocyte. Scale bar = 30 µm. (**B**) Percentage of oocyte viability depending on the estrous cycle phases in control and stressed female rats. The percentage of live oocytes decreased in stressed females in all estrous cycle phases. Data are shown as Mean ± SEM. * Significant difference vs. control when *p* < 0.05. The total number of evaluated oocytes was proestrus (control *n* = 185; stress *n* = 191), estrus (control *n* = 48; stress *n* = 46), and diestrus (control *n* = 147; stress *n* = 144).

**Figure 3 animals-11-02478-f003:**
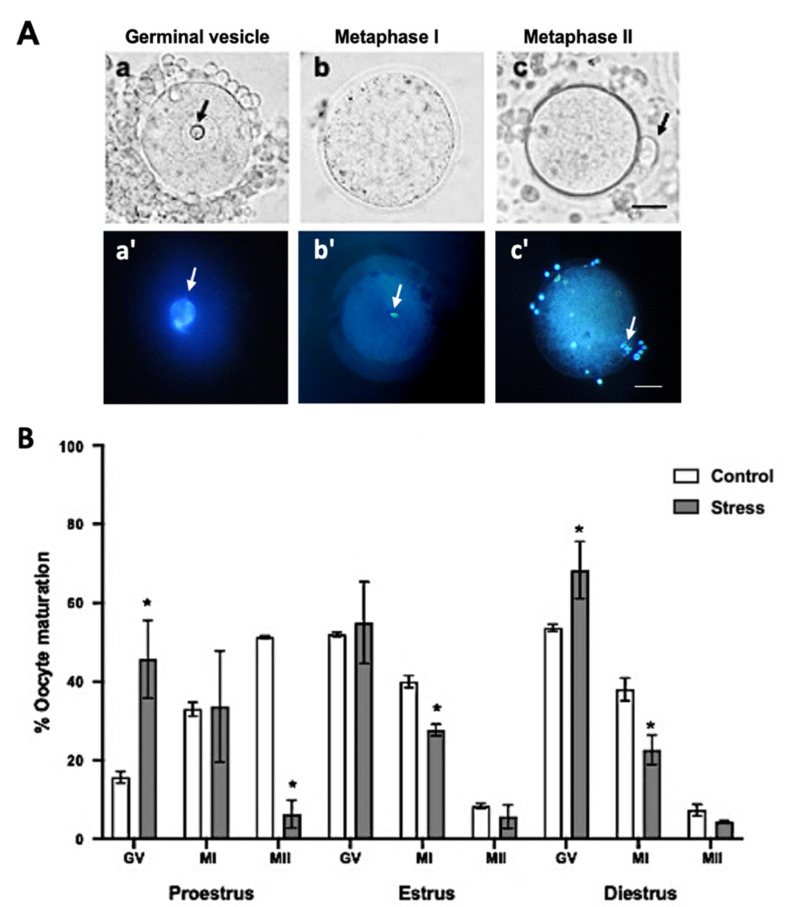
Oocyte maturation stages depending on the estrous cycle phases in control and stressed adult female rats. (**A**) Representative images of oocyte maturation stages with bright field (**a**–**c**) and Hoechst stain (**a’**–**c’**) evaluation: (**a**,**a’**) germinal vesicle (GV), the arrows indicate nucleolus, (**b**,**b’**) metaphase I (MI), the arrows indicate the metaphase and (**c**,**c’**) metaphase II (MII), the arrows indicate first polar body. Scale bar = 30 µm. (**B**) Percentage of oocyte maturation depending on the estrous cycle phases in control and stressed female rats. The percentage of GV oocytes increased in stressed females in the proestrus and diestrus phases. MI oocytes decreased in stress females in estrus and diestrus. Oocyte maturation (MII-stage) decreased in stressed females in proestrus. Data are shown as Mean ± SEM. * Significant difference vs. control when *p* < 0.05. The total number of evaluated oocytes was proestrus (control *n* = 185; stress *n* = 191), estrus (control *n* = 48; stress *n* = 46), and diestrus (control *n* = 147; stress *n* = 144).

**Figure 4 animals-11-02478-f004:**
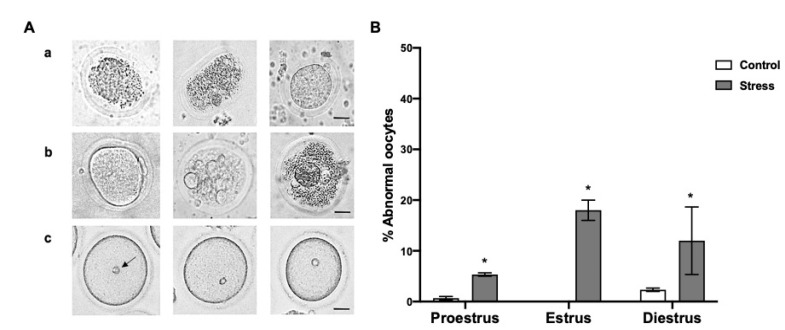
Abnormal oocytes depending on the estrous cycle phases in control and stressed adult female rats. (**A**) Representative images of abnormal oocytes: (**a**) oocytes with deformed zona pellucida and retracted or granular cytoplasm, (**b**) oocytes with abnormal zona pellucida and plasma membrane, and retracted, granular, and vacuolated cytoplasm, without spherical shape, (**c**) eccentric germinal vesicle (GV) position in the cytoplasm, the arrow indicates normal GV. Scale bar = 30 µm. (**B**) Percentage of abnormal oocytes depending on the estrous cycle phases in control and stressed female rats. The percentage of abnormal oocytes increased in stressed females in all estrous cycle stages. Data are shown as Mean ± SEM. * Significant difference vs. control group when *p* < 0.05. The total number of evaluated oocytes was proestrus (control *n* = 185; stress *n* = 191), estrus (control *n* = 48; stress *n* = 46), and diestrus (control *n* = 147; stress *n* = 144).

**Figure 5 animals-11-02478-f005:**
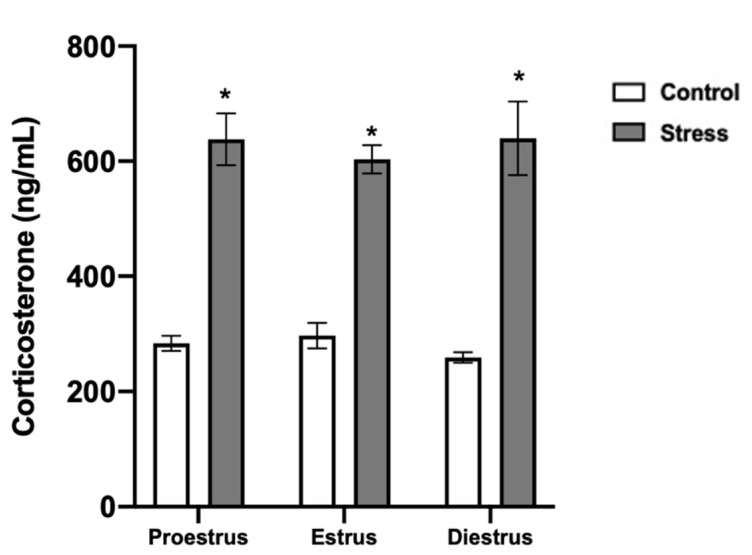
Corticosterone serum levels in control and stressed females. Corticosterone increased in stressed females in all estrous cycle phases. Data are shown as Mean ± SEM. * Significant difference vs. control when *p* < 0.05. Proestrus (*n* = 15), estrus (*n* = 15), and diestrus (*n* = 15).

## Data Availability

The data presented in this study are available on request from the corresponding author. The data are not publicly available due to the analysis program used.

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
