# Peer review of "Chronic Stress Detrimentally Affects In Vivo Maturation in Rat Oocytes and Oocyte Viability at All Phases of the Estrous Cycle"

_animals, 2021, doi:10.3390/ani11092478_

Round 1

Reviewer 1 Report

Stress provoke physical or physiological disorders. Many studies show a significant relationship between stress and deranged reproductive process in women. Stress can adversely alter the functionality of the hypothalamic-pituitary-gonadal (HPG) axis. Stress has been reported to cause subfertility in patients undergoing in-vitro fertilization (IVF). Therefore, it can be assumed that stress is closely related to poor IVF outcomes. The over-activation of the hypothalamic-pituitary-adrenal (HPA) axis and subsequently increase cortisol (in humans) or corticosterone (in rodents) secretion could strongly modifies not only the normal axis controlling the ovary but also the autonomic control of the ovary that is hyper activated by stress.

There are no studies evaluating the effects of chronic stress on oocytes collected in the different phases of the estrous cycle, without any exogenous hormonal stimulation. Therefore, the present study evaluated the effect of chronic stress on the viability, integrity, and in-vivo maturation of rat oocytes during three estrous cycle phases, including proestrus, estrus, and diestrus.

The stressful stimuli used was chronic stress by cold water immersion and studied oocyte viability,  integrity, and maturation. Female rats were stressed for 30 consecutive days, and vaginal  cytology was evaluated before being sacrificed. Oocyte viability, integrity, and maturation, as well as estrous cyclicity and serum corticosterone, were assessed at the end of stress exposure. Experiments were carefully done and results clearly presented.

The techniques used are sufficient to get good correlations. The work could be more strong with the plasma levels of steroidal hormones. Corticosterone is important to demonstrate the efficacy of the stress paradigm but the ovary function could be better explained with steroids, especially if there are clear changes in the estrous cycle regularity.

The conclusion shown by the authors however are well substantiated. The study showed that chronic stress affects oocyte developmental potential in the estrous cycle phases. In addition, chronic stress disrupted estrous cyclicity, decreased oocyte viability, and increased abnormal oocyte production in all estrous cycle phases, which resulted in reduced oocyte maturation during proestrus.
The results could be important for IVF procedures.

Suggestion: If the authors have the plasma levels of ovarian steroids, could increase the impact of the study

Reviewer 2 Report

This paper provides insights into the underpinning biological processes to reduced fertility associated with chronic stress. The data is clear and experiments well described. I have some suggestions for minor edits. I also think the manuscript would benefit from a native English speaker checking the sentence structure in places. Not sure I agree with the emphasis on all the different estrus phases other than the fact that after 30 days, all oocytes are adversely affected. Would be worth mentioning in discussion somewhere that a natural follow up experiment to this would be to fertilise a subsection of the oocytes and assess developmental potential.

Title: Slightly confusing grammatically. Suggest to alter to something like " Chronic stress detrimentally affects in vivo maturation in rat oocytes and oocyte viability at all phases of the estrus cycle"

Line 18: Frailty is not the appropriate word here in this context (line 44 also)

Line 26 and 36- Sentence is not correctly structured- adjustment needed

Line 68-Sentence not correctly structured.

Section 2.6. Query over sampling. You mention n=45 rats in each group but describe findings for only 35 in the stress group. What happened to the other 10 and how many were in each estrus phase in the control rats?

Section 3.2: results are repeated. Only one sentence needed here please condense to avoid repetition.

Section 4.2: Can you speculate on the mechanisms by which the changed HPA axis/kisspeptin levels you discuss translate to decreased oocyte viability as noted by the MTT stain i.e reduced metabolic activity?

Reviewer 3 Report

I find the aim of the study, the introduction and discussion very interesting, however from the point of view of materials and methods, experimental design and results, my opinion is that the manuscript would need to take into account other aspects not mentioned in the writing.

Summary

I find the aim of the study, the introduction and discussion very interesting, however from the point of view of materials and methods, experimental design and results, my opinion is that the manuscript would need to take into account other aspects not mentioned in the writing.

  1. What was the reason for not including the metestrus phase in the study?

  1. Could the authors clarify how many replicates were carried out in the experiment or if, on the contrary, all the biological samples of the experimental group were subjected to cold stress during the same period of 30 days?

  1. Why does it appears that oocyte recovery rate was much lower during estrus than in the other two phases of the estrous cycle studied?

  1. Why TL-HEPES-PVA medium was at 24 °C (L147)?

  1. Why did the authors only select oocytes with a diameter of 70 µm (L152)?.

  1. How long were COCs kept in the TCM 199 + 0.1% hyaluronidase medium? (L154-155)

  1. The sequence of analysis carried out with COCs and oocytes is not clearly explained in the writing. What was first oocyte denudation or evaluation of oocyte viability? (L152-155, L157)

  1. How long did it take to evaluate oocyte maturation since the COCs were collected? This factor seems to me critical for the interpretation of the results.

  1. Why did the authors not stain the oocytes with Hoechst  or DAPI to assess their nuclear maturation?

  1. Metestrus should be removed from Figure 2 (L200)

Round 2

Reviewer 3 Report

Authors should delete metestrus in Figure 1text (L200) M=Metestrus
